# Effect of Different Farming Practices on Lactic Acid Bacteria Content in Cow Milk

**DOI:** 10.3390/ani11020522

**Published:** 2021-02-17

**Authors:** Luciana Bava, Maddalena Zucali, Alberto Tamburini, Stefano Morandi, Milena Brasca

**Affiliations:** 1Department of Agricultural and Environmental Sciences, University of Milan, via Celoria 2, 20133 Milan, Italy; luciana.bava@unimi.it (L.B.); Alberto.tamburini@unimi.it (A.T.); 2National Research Council, Institute of Sciences of Food Production, via Celoria 2, 20133 Milan, Italy; stefano.morandi@ispa.cnr.it (S.M.); milena.brasca@ispa.cnr.it (M.B.)

**Keywords:** lactic acid bacteria, milk, farming practices, milk quality

## Abstract

**Simple Summary:**

Lactic acid bacteria (LAB) are the most important players to guarantee a correct cheesemaking process, and to define aroma profile and texture of the cheese. The natural prevalence of LAB in milk is variable, thus, the aim of the study was to identify the relationship between farm management practices, i.e., cow cleaning, bedding materials and management, ingredients in the feed ration and the presence of LAB and different other important groups of bacteria in cow bulk milk during different seasons. Information about farm management and milk bulk samples were collected in 62 dairy farms located in Po plain (Lombardy, Italy), most of them destined as milk for the production of Grana Padano Protected Denomination of Origin (PDO). Data from milk analyses and farm management practices were processed using multi-factor analyses in order to look for complex relations among variables, as in the farm environment. LAB content in milk did not result significantly different between seasons. Large farm dimension, high milk production and the application of a complete milking routine reduced microbial population in milk but promoted a high percentage trend of LAB on total bacteria count. The study underlined that the different management practices at the farm level could have an important effect on cheesemaking bacteria.

**Abstract:**

The natural load of lactic acid bacteria (LAB) in milk is the basis of the production of raw milk cheeses, such as Grana Padano PDO. In the last decades, improvements in livestock hygiene management resulted in bulk cow milk with less than 20,000 colony forming units (CFU) of bacterial count, unable to ensure a sufficient supply of LAB, with a negative impact on cheese quality. This study investigated the relations between farm management practices and prevalence of different groups of bacteria in cow milk. Sixty-two intensive dairy farms located in Lombardy (Italy) where involved, most of them destined as milk for the production of Grana Padano. Season had no significant effect on the content of most of the bacterial groups, except for coliforms. A strong relation among standard plate count (SPC) and other bacterial groups was evidenced. Cluster analysis showed that the most productive farms applied a complete milking routine and produced milk with the lowest value of SPC, the lowest count of the other bacteria, including LAB, but the highest LAB/SPC. The study suggests that complexity of farming practices can affect the microbial population of milk.

## 1. Introduction

In Europe, raw milk must meet the legal requirements for the bacterial load (≤100,000 colony forming units (CFU)/mL) and it must be free of pathogenic microorganisms when intended for human consumption (Regulation (EC) 853/2004), that means parameters closely related to milk spoilage and safety. On the other hand, the bacterial load, which constitutes an indicator of hygienic-sanitary quality, is not itself a parameter which guarantees the presence of lactic acid bacteria (LAB), which are the leading players in the production process of raw milk cheeses that represent productions of excellence in different countries [1].

LAB are classified as Gram-positive bacteria which include low Guanine + Cytosine (G + C) content as well as being acid tolerant, non-motile, non-spore forming and are rod- or cocci-shaped. The main function of LAB is to produce lactic acid, that is, the acidification of the food matrix. LAB have a lot of application as starter cultures in the food industry, with an enormous variety of fermented dairy products, meat, fish, fruit, vegetable and cereal products and feed production (silage forages) due to their acidifying capacity.

During cheesemaking, LAB are crucial for the formation of the curd, but they also contribute to cheese aroma and texture as they have endo- and exo-peptidases which are involved in the production of sapid molecules, and they generate precursors of aromatic compounds [2]. In addition, the non-starter lactic acid bacteria, mainly belonging to autochthonous microbiota, lead the ripening process of cheeses, which mainly explains the diversity among cheese types [3].

The natural load of LAB in raw milk is at the basis of the production of raw milk cheeses. In Italy, these types of cheeses are fundamental for the dairy sector: in 2019, 42% of total bovine milk cheeses were hard cheeses (namely Grana Padano PDO (Protected Denomination of Origin) and Parmigiano Reggiano PDO) obtained from raw milk, where the coagulation process is achieved with the use of rennet and natural whey starter [4]. However, the role of LAB is fundamental for the production of a great variety of traditional cheeses all over the world [5]. 

In Europe, in the last decades, improvements in livestock hygiene management resulted in the production of bulk cow milk with total bacterial count generally less than 20,000 CFU/mL [6]. This value, in itself positive, does not ensure a sufficient supply of lactic acid bacteria, with a negative impact on the quality of the cheese [7]. Recent studies highlighted that the milk microbial community at the farm level is affected by multiple factors that provide, on the whole, a peculiar distinctive microbiota composition for each farm, but further advances are needed to deepen the major drivers providing LAB prevalence and biodiversity in milk microbial population [8].

Some authors previously investigated the reasons for the reduction of LAB content. The authors of Reference [9] have shown that the microbial count and the balance between spoilage and useful cheese-making microorganisms can be influenced by a combination of milking practices (equipment, pre-milking and post-milking udder preparation). Verdier-Metz et al. [10] reported that teats’ washing with high concentrations of germicide or by using a paper towel reduce bacterial counts and also LAB. Moreover, Mallet et al. [1] concluded that maintaining freshly produced milk in cold storage, the development of udder-cleaning and teat-disinfecting procedures and monitoring the health status of dairy herds enhanced safety measures that reduced the levels of undesirable microorganisms (spoilage bacteria and potential pathogens), but this affected the entire natural microbiota spectrum. The same conclusion was reached in the study of Tormo et al. [2] in goat dairy farms, where the authors found that certain farming practices such as type of milking parlor and the presence of hay in the bedding area could enable to decrease the contamination of enterococci and promote the development of *L. lactis* in milk, improving technological and sensorial quality of lactic cheeses. Moreover, Cremonesi et al. [8] evidenced the influence of chlorine products’ usage in milking equipment on raw milk microbiota composition and biodiversity. It is worth considering that different studies [2,8,10] focused on the relation among bacterial diversity of milk obtained with phenotypic and genotypic methods and farm management practices, however only Mallet et al. [1] also considered the quantitative characterization of milk microbiota. 

In light of the above, the objective of this study was to investigate the relations between farm management practices, i.e., cow cleaning, bedding materials, ingredients in the feed ration and the prevalence of different types of microbial groups of bacteria in cow bulk milk during different seasons. 

## 2. Materials and Methods

### 2.1. Farm Characteristics

The study involved 62 dairy farms located in Po plain (Lombardy, Italy), where cows were bred in intensive confined livestock holdings, with no pasture, representing the most common farming system of Northern Italy. Most of the farms (54) produced milk destined for the production of Grana Padano PDO, so they conserved milk in bulk tanks at 9 °C following the PDO production specification rules. Twenty-six farms were visited once during the summer season and 36 farms were examined twice, also during the winter season, for a total of 98 visits. During the visit, information about the adopted management practices was collected during an interview with the farmer, in particular about herd size, feed ration composition, daily dry matter intake (DMI) and milk production.

The udder hygiene score (HS) was assessed in 34 farms: these farms were enrolled in a specific project focused on animals’ hygiene, and the assessments were performed through direct observation during milking, following the scheme proposed by Schreiner and Ruegg [11]. Udder hygiene score was expressed as percentage of udders with scores of 3 and 4, on a 4-point scale system, where score 1 indicates very clean skin while score 4 indicates skin completely covered with dirt.

### 2.2. Milk Analysis

Bulk milk was sampled directly from the tank after agitation, and milk was refrigerated for no more than 12 h at 4 or 9 °C for milk destined to Grana Padano PDO. Depending on the factory for which the milk was destined, bulk tanks contained the product of one or two consecutive milkings.

All samples were transported to the laboratory under refrigeration condition (4 °C) and processed for analyses no later than 12 h from the collection. 

Milk was analyzed for fat and protein content (MilkoScan FT6000; Foss Analytical A/S, Hillerod, Denmark) in the dairy service extension laboratory. Milk somatic cell count was detected using FOSSOMATIC TM 7 Electronic cell counter (FOSS, A/S, Hilleroed, Denmark) and the values obtained were log-transformed as Linear Score (LS), using the equation: LS = log_2_ ((somatic cell count, n/mL)/12,500) [12]. 

Standard plate count (SPC) was determined with Petrifilm AC plates (3M, Minneapolis, MN) incubated at 30 °C for 72 h, while psychotrophic bacteria were counted on AC plates incubated at 6.5 °C for 10 days according to ISO 6730: 2005 [13]. Coliforms were counted with 3M Petrifilm Coliform/*E. coli* Count Plates (3M) incubated for 24 h at 30 °C, while presumptive *Pseudomonas* counts were determined according to ISO/TS 11059:2009 [14] on *Pseudomonas* agar base (Oxoid Ltd., Basingstoke, UK) plates supplemented with Penicillin–Pimaricin (Biolife, Milan, Italy) incubated at 25 °C for 48 h. Lactic acid bacteria (LAB) were assessed using De Man Rogosa and Sharp (MRS) agar (Biolife) incubated at 30 °C for 72 h in anaerobic conditions (Anaerocult A, Merck Millipore, Darmstadt, Germany). The anaerobic spore-forming bacteria were enumerated by the Most Probable Number (MPN) method, as previously described by Zucali et al. [15]. Briefly, three 10-fold dilutions and five biological replicates were set up in reconstituted skimmed milk (10% *wt*/*v*) supplemented with yeast extract (1.0%), sodium lactate (3.36%), sodium acetate (1.0%), cysteine (0.2%) and sealed with vaseline/paraffin (1:1, *wt*/*wt*) plug. The inoculated milk was heat-treated at 80 °C for 10 min. Tubes exhibiting gas formation after incubation for 7 days at 37 °C were scored positive and results were calculated according to ISO/TS 7218:2013 [16]. 

All microbiological data were transformed into logarithmic with base 10 values and LAB content was also evaluated as percentage of the total bacterial load.

### 2.3. Statistical Analysis

The whole dataset was analyzed using SAS software (Version 9.4, 2012, SAS Institute Inc., Cary, NC, USA). Descriptive statistics were performed in order to describe dataset and variable distribution (Proc MEANS and Proc FREQ). Proc CORR were used in order to evaluate the correlation between different microorganisms tested.

Cluster analysis was performed (Proc CLUSTER) using the following variables: Individual milk production (kg/day), forage (% Dry Matter Intake-DMI), maize silage intake (Dry Matter-DM kg/day), hay intake (DM kg/day), lactating cows (n) and utilized agricultural area (ha). Cluster analysis identified 3 main clusters. Data were analyzed by Proc GLM to test differences between clustered farms. The effects of season (Winter, Summer) were tested using GLM analysis.

## 3. Results and Discussion

Most of the farms involved in the study (50) had cubicle sheds, while 12 of them housed cows on straw pack. Straw was the prevalent bedding material (35 vs. 13 that used sawdust). The herd dimension was 96.3 ± 56.9 lactating cows and utilized agricultural area was 44.6 ± 31.2 ha, on average (Table 1). Most of the farms included forages in lactating cows’ diet (58.1% ± 8.9% of forages on dry matter intake), mainly maize silage and hay. Individual milk production was on average 28.0 ± 5.2 kg/day and for most of the farms (54 out of 62), milk was destined for Grana Padano PDO cheese production. 

Linear Score (LS) from bulk milk samples resulted, on average, under the legal limit (400,000 cells/mL for high-quality milk, REG EU 853/2004 [17]), but the maximum level exceeded this threshold.

All farms performed milking twice a day. Regarding parlor type, 30 farms adopted herringbone, 14 pipeline and 12 parallel (3 observations were lost). Cleaning and disinfection of teat before milking (pre-dipping) was performed by 62% of the farms, while the use of post-dipping solution at the end of milking was confirmed as the operation most frequently used, as observed by Zucali et al. [18], and it was performed by 80.8% of the farms.

The pre-dipping procedure is widely recognized [19] as a useful procedure for decreasing bacterial contamination on the teat, on the other hand, it could reduce the load of LAB on the teat and consequently in the milk, as found by Verdier-Metz et al. [10].

The microbiological profile of bulk tank milk collected during summer and winter seasons is showed in Table 2.

Season had no significant effect for most of the bacterial groups, and significant differences were observed only for coliforms (2.73 vs. 1.65 Log_10_ CFU/mL) for summer and winter, respectively (*p* = 0.010). For anerobic spore-forming bacteria, a slight difference between seasons was observed (2.40 vs. 2.67 Log_10_ MPN/L for summer and winter, respectively; *p* = 0.064). Standard plate count showed, in both seasons, an average value below the legal limit of 100,000 CFU/mL (REG EU 853/2004). Zucali et al. [18] obtained similar counts during summer season for standard plate count (4.20 Log_10_ CFU/mL) and psychotrophic bacteria (3.84 Log_10_ CFU/mL), while Mallet et al. [1] reported lower values for standard plate count in spring compared to winter in a total of 260 raw milk samples from Bass-Normandie Region in France (4.10 vs. 5.60 Log_10_ CFU/mL).

In the present study, LAB count did not result significantly different between seasons, which suggests that LAB count in milk would depend more on management practices or environmental characteristics than on temperature and humidity variations. On the other hand, this result could underline that management practices are effective to reduce bacterial count also during more critical periods of the year.

A deepening of the distribution of LAB in milk samples in the two seasons is presented in Figure 1. Analyzing LAB content in the milk of the single farms in relation to the different seasons (Figure 1A), it is possible to notice a different distribution of samples in summer and winter. Summer data resulted with a shape similar to normal distribution. Considering LAB count expressed as percentage of SPC (Figure 1B), two groups of farms are distinguishable: a first group characterized by a very low content of LAB (less than 75%), where the highest level of LAB is in winter samples, and a second one characterized by a high content of LAB (always exceeding 75%) that reaches the highest percentages in summer. Thus, it is possible to conclude that during summer, there is a slight tendency toward an increase of milk LAB proportion. 

As expected, there was a strong and significant relation among SPC and other bacterial groups, except for with the anerobic spore-forming bacteria, whose presence is mainly related to the diet composition and cleanliness of the animals (Table 3), as reported by Zucali et al. [15]. The inverse correlation between the ratio of LAB and SPC and *Pseudomonas* content confirms what has already been shown by numerous studies, i.e., that refrigerated storage of milk favors the development of bacteria belonging to the genus *Pseudomonas* to the detriment of lactic acid bacteria [20].

Finally, all the farms included in the evaluation were classified, with the support of cluster analysis based on variables described in the previous section. Three clusters were obtained and in Table 4, the analysis of variance is shown for the most interesting variables. The clusters identified did not differ widely from each other; as already mentioned, all farms kept the herd in permanent confinement without pasture, all farms were located in the same area and were milk production specialist farms. Most of the farms produced milk for Grana Padano PDO cheese (81.4%, 85.7% and 85.2% for clusters 1, 2 and 3). In the first cluster, there are the largest farms, represented as number of lactating cows, and the most productive ones, represented as daily milk production, although not significant. The dairy cows’ ration included a higher quantity of maize silage and lower percentage of forages compared to the other two groups of farms, although not significant. Most of the farms applied a complete milking routine with pre-dipping (16 yes vs. 9 no) and post-dipping (79%) (Figure 2A). The udder cleanliness was very good, with the lowest percentage of dirty udders (Figure 2B). In the farms belonging to cluster 1, animals were bred in cubicles (81%), and most farms used straw as bedding (26 farms), as shown in Figure 3. Compared with the other two clusters, milk produced by farms of cluster 1 was characterized by the lowest and significantly different values of SPC in milk, and this result is consistent with Zucali et al. [18], who showed a positive effect of application of two or more milking operations (fore-stripping, pre-dipping, post-dipping) on SPC milk content, and with the conclusion of Mallet et al. [1], who showed that the pre-dipping was associated with lower levels of SPC. Also, the other microbial groups considered, LAB included, registered the lowest values in comparison to the other clusters; indeed, the differences were not significant, and this is consistent with the conclusion of Verdier-Metz et al. [10]. 

Moreover, it is interesting to note that, although not significant, in this group, the highest percentage of LAB contribution to SPC was observed. It is particularly remarkable to note the lowest and significant value of anerobic spore-forming bacteria count in this group of farms, despite the high inclusion of silages in the diet. Vissers et al. [21] demonstrated that silage contamination is the main source of spore milk content, but recent studies provided evidence that the inclusion in the diet of high-quality maize silage in terms of pH and acid content does not cause any increase of anerobic spore-forming bacteria level in milk [18]. Moreover, the greater cow cleanliness together with the application of a complete milking routine characterizing these farms can explain the reduced spore content in milk [18].

Farms in cluster 2 showed the lowest individual milk production, and microbial count had intermediate values compared to the other two clusters. In cluster 2, the LAB/SPC ratio had the lowest value. The farms included in this cluster were characterized by lower frequency application of post-dipping and more frequent use of straw as bedding compared with cluster 1 (Figure 2 and Figure 3).

The highest content of SPC, psychotrophs, coliforms and anerobic spore-forming bacteria were observed in the milk of cluster 3. The milk of this group of farms was also characterized by the highest level of LAB, but no significant differences were highlighted. These farms were characterized by low dimension of herd, use of pre-dipping in about half of the farms and lower use of post-dipping compared with other clusters (Figure 2A). Dairy cows in these farms also showed the worst udder hygiene (Figure 2B); although the type of housing was similar to farms of cluster 1, the difference of udder hygiene was probably connected to the management of bedding, such as frequency of clean material and manure removal. The result in terms of microbiological load of milk was consistent with observations of Zucali et al. [15], where non-clean cows produced milk with high bacterial count, and Elmoslemany et al. [22], who demonstrated that the amount of dirt on teats before pre-milking udder preparation was positively associated with SPC and psychotrophic bacterial count in milk. 

## 4. Conclusions

This study, conducted in intensive dairy farms in North Italy, aimed to explore the relationship among the different management practices applied and the microbiological quality of milk with particular attention to LAB, the most important group of microorganisms for the cheese-making process, with a fundamental role for raw milk cheese production. The results underline a strong positive relationship among SPC and most of the microbial groups considered in the study, suggesting that milk with a very low mesophilic bacteria count could reduce its potential to transform milk into raw milk cheese. Moreover, cluster analysis showed that the management system applied was not much different between farms, all belonging to intensive systems. The high producing farms showed the lowest value of SPC, along with lower levels of undesirable microbial groups, such as anerobic spore-forming bacteria and, although to a lesser extent, coliforms and psychotrophs, while maintaining a nearly unchanged LAB content. Milk from these farms was characterized by the best ratio between LAB and SPC, indeed not significantly different from other clusters, suggesting that the factors affecting this parameter are many and complex. The indications obtain from this study could be useful to cheese factory operators in order to select farms with specific management that produce proper raw milk.

## Figures and Tables

**Figure 1 animals-11-00522-f001:**
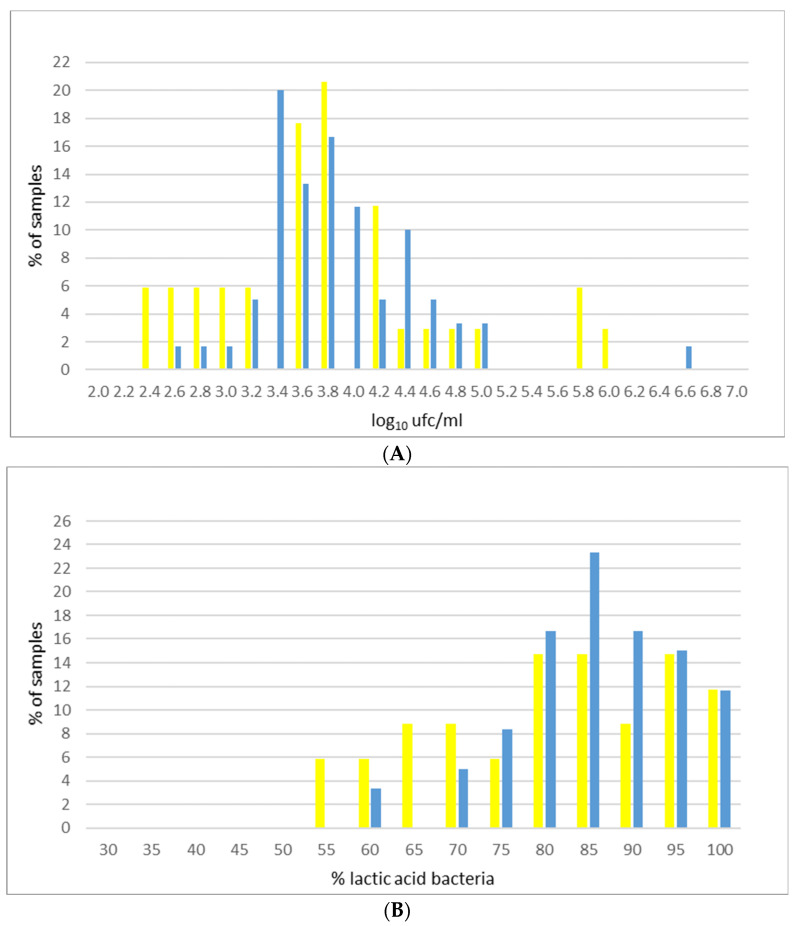
Distribution of LAB (**A**) and percentage of LAB on SPC (**B**) in bulk tank milk samples (blue bars = winter; yellow bars = summer).

**Figure 2 animals-11-00522-f002:**
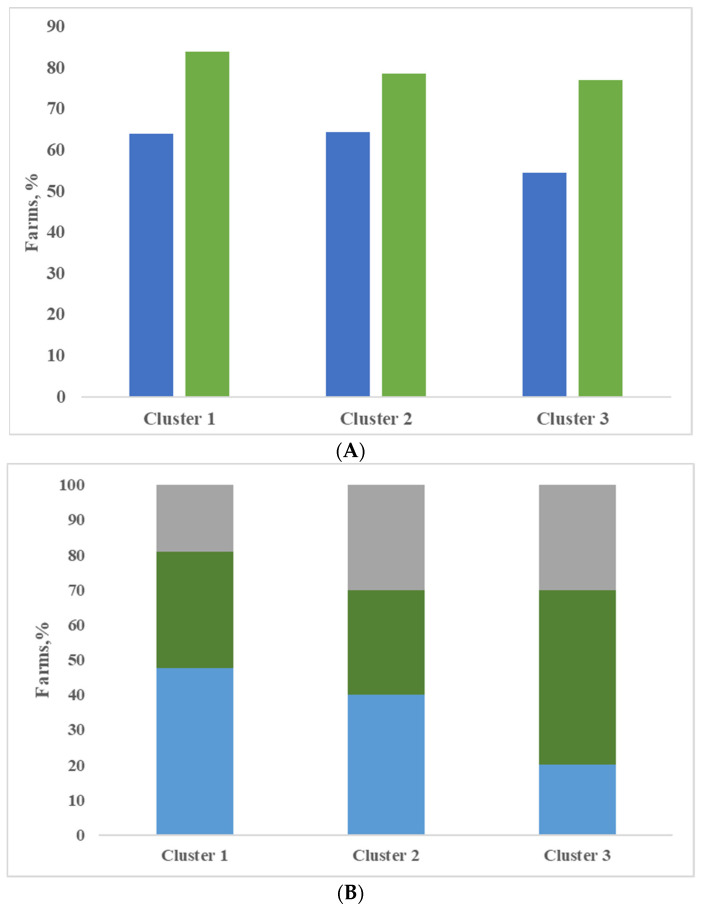
Execution of pre- and post-dipping during milking routine (**A**) and udder hygiene score (HS) (**B**) expressed as percentage of 3 + 4 score in the three clusters (A: blue bar = pre-dipping; green bar = post-dipping; B = grey bar = HS > 40%; green bar = HS 20–40%; blue bar = HS < 20%).

**Figure 3 animals-11-00522-f003:**
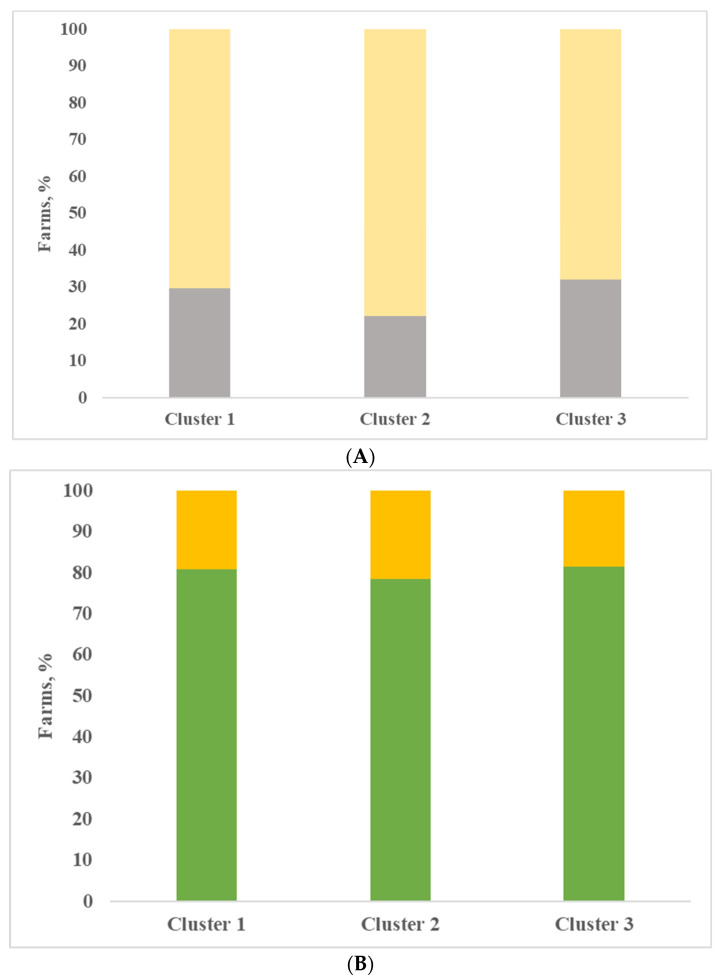
Type of bedding (**A**) and housing (**B**) in the three clusters (A: gray bar = sawdust; yellow bar =straw; B: green bar = cubicle; yellow bar = straw pack).

**Table 1 animals-11-00522-t001:** Farm and diet characteristics.

Variable	Unit	N	Mean	SD	Minimum	Maximum
Utilized agricultural area	ha	85	44.6	31.2	8.0	159
Lactating cows	n	98	96.3	56.9	15.0	270
Udder hygiene score (3–4)	%	42	29.7	25.9	0.0	100
Milk production						
Individual milk production	kg/day	98	28.0	5.2	15.1	40.3
Fat	%	98	3.65	0.29	3.02	4.39
Protein	%	98	3.75	0.26	3.29	4.29
Linear Score		80	3.52	0.84	1.41	5.35
Lactating cow’s diet composition						
Feed intake	DM kg/day *	94	21.7	2.53	14.1	28.5
Maize silage intake	DM kg/day *	88	6.5	2.63		10.1
Hay intake	DM kg /day *	87	3.7	2.78		12.0
Forage	% of DMI ^	90	58.1	8.9	39.7	86.5
Maize silage	% of DMI ^	94	29.5	12.7		47.4
Hay	% of DMI ^	94	16.4	12.7		46.6

* DM: dry matter; ^ DMI: dry matter intake.

**Table 2 animals-11-00522-t002:** Microbiological profile of bulk tank milk samples (least square means).

Bacterial Group	Unit	Summer	Winter	SEM ^a^	*p*
Standard plate count (SPC)	Log_10_ CFU ^b^/mL	4.56	4.46	0.18	0.636
Lactic acid bacteria (LAB)	Log_10_ CFU/mL	3.90	3.76	0.17	0.517
Coliforms	Log_10_ CFU/mL	2.73	1.65	0.32	0.010
*Pseudomonas* spp.	Log_10_ CFU/mL	3.52	3.93	0.27	0.206
Psychotrophic bacteria	Log_10_ CFU/mL	4.36	4.58	0.22	0.490
Anerobic spore count	Log_10_ MPN ^c^/L	2.40	2.67	0.11	0.064
LAB/SPC	%	88.1	86.5	8.70	0.883

^a^ Standard error of the mean, ^b^ Colony forming units, ^c^ Most probable number.

**Table 3 animals-11-00522-t003:** Correlation coefficients among different microbial groups in milk as Log_10_ CFU/mL (different color of cells meaning different level of significance: red *p* < 0.0001; blue *p* < 0.05; white *p* = Not significant (NS).

Microbial Groups	Anaerobic Spore Count	SPC	Coliforms	*Pseudomonas* spp.	Psychotrophic Bacteria	LAB	LAB/SPC
**Anerobic spore count**	1						

**SPC**	0.055	1					

**Coliforms**	−0.122	0.655	1				

***Pseudomonas* spp.**	−0.058	0.622	0.451	1			

**Psychotrophic bacteria**	0.038	0.655	0.494	0.672	1		

**LAB**	0.159	0.604	0.412	0.243	0.458	1	

**LAB/SPC**	−0.082	−0.498	−0.190	−0.273	−0.122	0.158	1

**Table 4 animals-11-00522-t004:** Analysis of variance between clusters (CL) (LS means value).

Variable Name	Unit	Cluster 1	Cluster 2	Cluster 3	SEM	*p*	CL1 vs. CL2	CL1 vs. CL3	CL2 vs. CL3
Observation	n	48	28	27					
Utilized agricultural area	ha	44.3	43.7	47.0	6.790	0.929	0.949	0.754	0.720
Lactating cows	n	103	94.1	90.8	11.75	0.663	0.525	0.400	0.834
Milk production									
Individual milk production	kg/day	29.6	26.8	27.9	1.040	0.073	0.025	0.189	0.424
Fat	%	3.68	3.61	3.6	0.060	0.502	0.330	0.326	0.970
Protein	%	3.71	3.85	3.76	0.050	0.079	0.025	0.417	0.204
LS	Log_10_ CFU/mL	3.59	3.26	3.33	0.170	0.270	0.144	0.233	0.791
Lactating cows’ diet composition									
Maize silage	% DM intake	30.5	27.7	29.3	2.550	0.675	0.377	0.724	0.636
Hay	% DM intake	16.3	18.7	15.4	2.630	0.628	0.460	0.798	0.362
Forage	% intake	57	59.3	59.0	1.920	0.543	0.326	0.395	0.928
Bacteria count									
Standard plate count (SPC)	Log_10_ CFU/mL	4.35	4.69	4.88	0.170	0.034	0.099	0.013	0.392
Lactic acid bacteria (LAB)	Log_10_ CFU/mL	3.72	3.8	4.09	0.150	0.143	0.670	0.053	0.157
Coliforms	Log_10_ CFU/mL	2.09	2.43	2.88	0.290	0.096	0.328	0.031	0.240
*Pseudomonas* spp.	Log_10_ CFU/mL	3.54	4.04	3.98	0.210	0.110	0.058	0.102	0.856
Psychotrophic bacteria	Log_10_ CFU/mL	4.16	4.48	4.86	0.240	0.103	0.306	0.034	0.256
Anerobic spore count	Log_10_ MPN/mL	2.26	2.38	2.67	0.130	0.041	0.439	0.012	0.099
LAB/SPC	%	91.1	80.8	83.7	6.780	0.417	0.211	0.381	0.758

Numbers in bold have *p* < 0.05.

## Data Availability

Not applicable.

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
