# Peer review of "Effect of Different Farming Practices on Lactic Acid Bacteria Content in Cow Milk"

_animals, 2021, doi:10.3390/ani11020522_

Round 1
Reviewer 1 Report
Recommendation: Minor Revision
Line 2: Title can be changed…………: effect of different farming practices
Line 47: Need citation
Line 51 to 54: Please rearrange the sentence, For example: LAB have a lot of application as starters cultures in the food industry with an enormous variety of fermented dairy products, meat, fish, fruit, vegetable and cereal products and feed production (silage forages) due to their acidifying capacity.
Line 1110 t0 119: You said the study involved 62 farms but you visited 98 farms? Please clear it.
Line 120: Why did you measure the hygiene score in 34 farms not others? Please explain it
Line 155: Why did you transformed your data in logarithmic scale? Are your data not normally distributed?
Line 164: 3 groups of farms were identified using 164 proc CLUSTER analysis? Which farm criteria did you consider to cluster them? Here is not clear?
Line 169 to 171: Please consider these lines in Materials and Methods Part. These lines are unnecessary in results and discussion part.
Line 182: Not necessary to keep the Table 1. Just you can mention your average value as descriptive form in the results and discussion section.
Line 274:….in the milk of Cluster 3
Line 291 to 292: In this page please mention the Figure number and title. Also remove the background line from all figures.
Table 4: you can put CL1 vs CL2; CL1 vs CL3; CL2 vs CL3 instead of 1 vs 2; 1 vs 3 ; 2 vs 3.
Line 298: ……among the different management practices
Author Response
Recommendation: Minor Revision
Line 2: Title can be changed…………: effect of different farming practices
AU: Thank you for the suggestion the new title will be ‘Effect of different farming practices on Lactic acid bacteria content in cow milk’
Line 47: Need citation
AU: we added a citation as requested
Line 51 to 54: Please rearrange the sentence, For example: LAB have a lot of application as starters cultures in the food industry with an enormous variety of fermented dairy products, meat, fish, fruit, vegetable and cereal products and feed production (silage forages) due to their acidifying capacity.
AU:We modified the sentence as suggested.
Line 1110 t0 119: You said the study involved 62 farms but you visited 98 farms? Please clear it.
AU:We modified the sentence in order to be more understandable
Line 120: Why did you measure the hygiene score in 34 farms not others? Please explain it
AU:We modified the sentence in order to be more understandable
Line 155: Why did you transformed your data in logarithmic scale? Are your data not normally distributed?
AU: Microbiological data are always not normally distributed. We use log10 to transform data in a normal distribution.
Line 164: 3 groups of farms were identified using 164 proc CLUSTER analysis? Which farm criteria did you consider to cluster them? Here is not clear?
AU:We modified the sentence in order to be more understandable, because we used CA on 62 observations using 6 numeric variables (as described): Individual milk production (kg/day); Forage (% DMI); Maize silage intake (DM kg/d); Hay intake (DM kg /d); lactating cows (n); Utilized agricultural area (ha)
Line 169 to 171: Please consider these lines in Materials and Methods Part. These lines are unnecessary in results and discussion part.
AU: we deleted the sentence in the result and discussion part, because, as underline, it was already explained in the material and method section
Line 182: Not necessary to keep the Table 1. Just you can mention your average value as descriptive form in the results and discussion section.
AU: we prefer to keep table 1, we modified the format in order to make it more readable
Line 274:….in the milk of Cluster 3
AU:We modified as suggested
Line 291 to 292: In this page please mention the Figure number and title. Also remove the background line from all figures.
AU: Figure number and title is reported at lines 289-290
Table 4: you can put CL1 vs CL2; CL1 vs CL3; CL2 vs CL3 instead of 1 vs 2; 1 vs 3 ; 2 vs 3.
AU: we modified as suggested
Line 298: ……among the different management practices
AU: we modified as suggested
Reviewer 2 Report
The study by Bava et al. aimed at investigating the relations between farm management practices and prevalence of different groups of bacteria, especially lactic acid bacteria, in cow milk. Sixty-two intensive dairy farms located in Lombardy (Italy) where involved, of which 36 were visited both in summer and in winter. The main microbial groups in milk were enumerated on culture media and the results were analayzed according to farm management practices.
General comments :
I recognize the efforts of the authors to conduct a large-scale study on a large number of farms. However, the authors should be more careful in their conclusions. Overall, the farms studied are all intensive in nature and appear to apply very similar practices. The authors repeatedly point out differences that are not statistically significant. None of the variables studied can explain, on its own and with a supporting statistical test, the differences in the level of SPC, coliforms, psychrotrophs and anaerobic spores between the farms of clusters 1 and 3. This is a conclusion that should be emphasized. On the other hand, the LAB / SPC ratios are not different between the clusters, contrary to what the authors argue.
Overall, the article would benefit from a proofreading by an English-speaking proofreader.
Specific comments :
Abstract :
Line 25 : « enable » should read « unable » ?
Introduction
Line 61-65 : The paragraph on the health effects of LAB seems out of the scope of the article.
Lines 60, 66 : LABs play a major role in the production of raw milk cheeses, but they are not the only ones.
Materials and methods
Line 120 : Was the HS measurement taken before the cows entered the milking parlor? thank you for specifying.
Results
Line 206 : add « count » after LAB
Line 215 : please explain what you mean by "regular".
Line 221: the term "content" is confusing: rather use "percentage" or "proportion"
Lines 220-221 : Do you have any statistical tests that show that the distribution of levels (Figure 1A) and proportion (Figure 1B) of LAB is different in summer and winter?
Table 3:
Please specify what the indicated values correspond to: are they correlation coefficients?
Lines 235-256 : None of the variables describing the farming practices presented in Table 4 are significantly different between clusters 1 and 3. With regard to milking practices, udder cleanliness (Figure 2), type of bedding and animal housing (Figure 3), it is difficult to assess the differences because no statistical test is presented. It might be wise to search for farming practices that might explain the differences in udder cleanliness while the types of bedding or housing are very similar.
Author Response
The study by Bava et al. aimed at investigating the relations between farm management practices and prevalence of different groups of bacteria, especially lactic acid bacteria, in cow milk. Sixty-two intensive dairy farms located in Lombardy (Italy) where involved, of which 36 were visited both in summer and in winter. The main microbial groups in milk were enumerated on culture media and the results were analayzed according to farm management practices.
General comments :
I recognize the efforts of the authors to conduct a large-scale study on a large number of farms. However, the authors should be more careful in their conclusions. Overall, the farms studied are all intensive in nature and appear to apply very similar practices. The authors repeatedly point out differences that are not statistically significant. None of the variables studied can explain, on its own and with a supporting statistical test, the differences in the level of SPC, coliforms, psychrotrophs and anaerobic spores between the farms of clusters 1 and 3. This is a conclusion that should be emphasized. On the other hand, the LAB / SPC ratios are not different between the clusters, contrary to what the authors argue.
AU: see comments below
Overall, the article would benefit from a proofreading by an English-speaking proofreader.
AU: we check English language and we modified were needed.
Specific comments :
Abstract :
Line 25 : « enable » should read « unable » ?
AU: we modified in the text as suggested
Introduction
Line 61-65 : The paragraph on the health effects of LAB seems out of the scope of the article.
AU: we decided to delete this sentence as suggested.
Lines 60, 66 : LABs play a major role in the production of raw milk cheeses, but they are not the only ones.
AU: we agree with this comment, LABs are not the only one which have a role in raw milk cheese production, on the other hand we believe that they are the main characters.
Materials and methods
Line 120 : Was the HS measurement taken before the cows entered the milking parlor? thank you for specifying.
AU: the HS assessment was performed during milking, when the cows stop in the milking parlour.
Results
Line 206 : add « count » after LAB
AU: we modified in the text as suggested
Line 215 : please explain what you mean by "regular".
AU: we modified the sentence in order to be more understandable
Line 221: the term "content" is confusing: rather use "percentage" or "proportion"
AU: we modified in the text as suggested, using ‘proportion’
Lines 220-221 : Do you have any statistical tests that show that the distribution of levels (Figure 1A) and proportion (Figure 1B) of LAB is different in summer and winter?
AU: thank you for the suggestion, we did not perform a specific statistical test on distributions. We statistically tested the differences among seasons using GLM procedure.
Table 3:
Please specify what the indicated values correspond to: are they correlation coefficients?
AU: we modified the title of the table as suggested
Lines 235-256 : None of the variables describing the farming practices presented in Table 4 are significantly different between clusters 1 and 3. With regard to milking practices, udder cleanliness (Figure 2), type of bedding and animal housing (Figure 3), it is difficult to assess the differences because no statistical test is presented. It might be wise to search for farming practices that might explain the differences in udder cleanliness while the types of bedding or housing are very similar.
AU: we thank the reviewer for the suggestion, unfortunately we don’t have reliable information on bedding management, as frequency of removal or frequency of renewal. These aspects are important but not easy to be get from farmers. Our database was composed by all intensive farms without pasture, so types of management were not drastically different. In a future study we think could be interesting to compare bacterial load and LAB content in different farming systems (intensive vs extensive).
We added a sentence in order to clarify this aspect.
Reviewer 3 Report
Dear Authors,
the manuscript deals with an interesting study on the effect of farming practices on Lactic acid bacteria content in cow milk. The paper is well written and considers an aspect which can play a key role in the production process of raw milk cheeses. Nevertheless, I have some concerns regarding statistical analisys and the shape of the tables and figures.
Statistical analisys
in chapter 2.3. I understand that to evaluate the microbiological profile you have only evaluated the season effect. Wouldn't it have been better to consider a model that also included the farm effect, which probably would have had an equal if not greater weight than the season? In light of these considerations, please explain your choices in more detail in satistical analysis.
Table and figures
The presentation of data in graphs and tables needs to be improved. Figure 1 is difficult to read because there are too many data condensed in a small space (I recommend splitting the figure). For tables suggestions see below.
Other comments
------------------------------------------------------------
Line
100 Please, insert Adduci et al. [xx]
321 remove space
325 year not in bold
330 write “LWT Food Sci. Technol.”
351 83
356 90
Table 1 - The items of the variables are not clearly readable and in some cases they are combined with the units of measure in the secund column (e.g. udder hygiene score, individual milk production ...). You could enlarge the space of the first column, shrink the others and use a smaller font size (in this Journal the tables can be edited with smaller font bodies, up to size 8)
Table 2 - See table 1
Table 3 - This matrix should also be reshaped
Table 4 – The second line of the header is incomplete, see also comments in table 1
Figure 1 - It is useless to start the x axis from the value 0. If it starts from 2.0 in 1A and from 50 in 1B you can enlarge the histograms thus making them more intelligible.
-------------------------------------------------------------------------
[XX] Adduci, F.; Elshafie, H.S.; Labella, C.; Musto, M.; Freschi, P.; Paolino, R.; Ragni, M.; Cosentino, C. Abatement of the clostridial load in the teats of lactating cows with lysozyme derived from donkey milk. J. Dairy Sci. 2019, 102, 6750–6755, https://doi.org/10.3168/jds.2019-16311.
Author Response
Dear Authors,
the manuscript deals with an interesting study on the effect of farming practices on Lactic acid bacteria content in cow milk. The paper is well written and considers an aspect which can play a key role in the production process of raw milk cheeses. Nevertheless, I have some concerns regarding statistical analisys and the shape of the tables and figures.
Statistical analisys
in chapter 2.3. I understand that to evaluate the microbiological profile you have only evaluated the season effect. Wouldn't it have been better to consider a model that also included the farm effect, which probably would have had an equal if not greater weight than the season? In light of these considerations, please explain your choices in more detail in satistical analysis.
AU: we did not use a “Farm Effect” because 26 farms were visited once during summer season and 36 farms were examined twice (also during winter season). In the Cluster Analysis we introduced variables linked to farm management, as lactating cows (n) or Utilized Agricultural Area (ha), and we used Forage (% DMI), Maize silage intake (DM kg/d) and Hay intake (DM kg /d) linked to feeding system in the farms.
Table and figures
The presentation of data in graphs and tables needs to be improved. Figure 1 is difficult to read because there are too many data condensed in a small space (I recommend splitting the figure). For tables suggestions see below.
AU: we improved all figures graphically. Tables view we think it is a conversion problem from word file to pdf one.
Other comments
------------------------------------------------------------
Line
100 Please, insert Adduci et al. [xx]
AU: thank you for your suggestion but we think that the references reported in this point of the paper are enough and complete. The paper suggested doesn’t give, in our opinion, additional information or comparisons.
321 remove space
AU: we modified in the text as suggested, we removed doi indication
325 year not in bold
AU: we modified in the text as suggested
330 write “LWT Food Sci. Technol.”
AU: we modified in the text as suggested
351 83
AU: we modified in the text as suggested
356 90
AU: we modified in the text as suggested
Table 1 - The items of the variables are not clearly readable and in some cases they are combined with the units of measure in the secund column (e.g. udder hygiene score, individual milk production ...). You could enlarge the space of the first column, shrink the others and use a smaller font size (in this Journal the tables can be edited with smaller font bodies, up to size 8)
AU: We modified the table as suggested, in the word (.doc) file that we uploaded the view of the table was different
Table 2 - See table 1
AU: we modified as suggested
Table 3 - This matrix should also be reshaped
AU: we modified as suggested
Table 4 – The second line of the header is incomplete, see also comments in table 1
AU: also in this case there is a visualization problem, probably in the conversion from .doc to pdf. The second line is the number of observation which does not have significant values.
Figure 1 - It is useless to start the x axis from the value 0. If it starts from 2.0 in 1A and from 50 in 1B you can enlarge the histograms thus making them more intelligible.
AU: we modified graphically figure 1A and 1B, as suggested also by another reviewer, we believe that now it can be more clear
-------------------------------------------------------------------------
[XX] Adduci, F.; Elshafie, H.S.; Labella, C.; Musto, M.; Freschi, P.; Paolino, R.; Ragni, M.; Cosentino, C. Abatement of the clostridial load in the teats of lactating cows with lysozyme derived from donkey milk. J. Dairy Sci. 2019, 102, 6750–6755, https://doi.org/10.3168/jds.2019-16311.
Reviewer 4 Report
These studies provide a comprehensive overview of the effect of various factors on milk quality. This is a very important problem in milk processing.
Detailed comments
Line 42 - Regulation (EC) 178/2002 of the European Parliament and of the Council of 28 January 2002 laying down the general principles and requirements of food law, establishing the European Food Safety Authority and laying down procedures in matters of food safety. The regulations you write about are written in Regulation (EC) no 853/2004. Correct.
Line 113-114 You write about conserved bulk milk. What method?
Line 179-180 You write that:
„Linear Score (LS) from bulk milk samples resulted, on average, under legal limit (300 000 cells/mL for high quality milk, REG EU 853/2004), but the maximum 180 level exceeded this threshold.”
It is written exactly in the Regulation (no 853/2004 of the European Parliament and of the Council of 29 April 2004 laying down specific hygiene rules for food of animal origin):
Food business operators must initiate procedures to ensure that raw milk meets the following criteria:
- for raw cows' milk: plate count at 30 oC (per ml) ≤ 100 000 ; somatic cell count (per ml) ≤ 400 000
Food business operators manufacturing dairy products must initiate procedures to ensure that, immediately before being heat treated and if its period of acceptance specified in the HACCP-based procedures is exceeded:
- raw cows’ milk used to prepare dairy products has a plate count at 30°C of less than 300 000 per ml;
Whether the milk was intended for the production of cheese without heat treatment? Have they been pasteurized?
There is no answer in the conclusions, did the tested milk meet the requirements of Regulation 853/2004? Have exceedances been demonstrated?
Author Response
These studies provide a comprehensive overview of the effect of various factors on milk quality. This is a very important problem in milk processing.
Detailed comments
Line 42 - Regulation (EC) 178/2002 of the European Parliament and of the Council of 28 January 2002 laying down the general principles and requirements of food law, establishing the European Food Safety Authority and laying down procedures in matters of food safety. The regulations you write about are written in Regulation (EC) no 853/2004. Correct.
AU: we modified as suggested
Line 113-114 You write about conserved bulk milk. What method?
AU: we modified the sentence in order to clarify that milk was conserved in bulk tank
Line 179-180 You write that:
„Linear Score (LS) from bulk milk samples resulted, on average, under legal limit (300 000 cells/mL for high quality milk, REG EU 853/2004), but the maximum level exceeded this threshold.”
It is written exactly in the Regulation (no 853/2004 of the European Parliament and of the Council of 29 April 2004 laying down specific hygiene rules for food of animal origin):
Food business operators must initiate procedures to ensure that raw milk meets the following criteria:
- for raw cows' milk: plate count at 30 oC (per ml) ≤ 100 000 ; somatic cell count (per ml) ≤ 400 000
Food business operators manufacturing dairy products must initiate procedures to ensure that, immediately before being heat treated and if its period of acceptance specified in the HACCP-based procedures is exceeded:
- raw cows’ milk used to prepare dairy products has a plate count at 30°C of less than 300 000 per ml;
Whether the milk was intended for the production of cheese without heat treatment? Have they been pasteurized?
There is no answer in the conclusions, did the tested milk meet the requirements of Regulation 853/2004? Have exceedances been demonstrated?
AU: we modified the sentence. Grana Padano cheese is made using raw milk so the microbiological quality of this is crucial.
Round 2
Reviewer 2 Report
The authors have made satisfactory changes, except the simple summary and the summary that should be revised to be in line with the new version of the conclusion.
Author Response
Thank you for your suggestion. We revised the simple summary and the abstract as suggested.
Reviewer 3 Report
Dear Authors,
the suggestions were properly received.
Best regards
Author Response
Thank you